# Understanding Employees’ Attitudes and Awareness of Code of Ethics and Associated Factors: A Cross-Sectional Survey at a Public Tertiary Hospital in Croatia

**DOI:** 10.3390/healthcare13172131

**Published:** 2025-08-27

**Authors:** Zrinka Hrgović, Jure Krstulović, Ante Tavra, Ante Krešo, Franko Batinović, Ljubo Znaor, Ana Marušić

**Affiliations:** 1School of Medicine, University of Split, Šoltanska 2A, 21000 Split, Croatia; jure.krstulovic@mefst.hr (J.K.); ante.tavra@mefst.hr (A.T.); 2Department of Family Medicine, Split-Dalmatia Health Center, Kavanjinova 2, 21000 Split, Croatia; 3Department of Health Care Quality, University Hospital of Split, Spinčićeva 1, 21000 Split, Croatia; 4Department of Surgery, University Hospital of Split, Spinčićeva 1, 21000 Split, Croatia; 5Department of Pediatric Disease, Division of Haematology, Oncology, Clinical Immunology and Genetics, University Hospital of Split, Spinčićeva 1, 21000 Split, Croatia; 6Department of Ophthalmology, University Hospital of Split, Spinčićeva 1, 21000 Split, Croatia; ante.kreso@mefst.hr (A.K.); ljubo.znaor@mefst.hr (L.Z.); 7Department of Otorhinolaryngology, University Hospital of Split, Spinčićeva 1, 21000 Split, Croatia; fbatinovic@kbsplit.hr; 8Department of Ophthalmology, University of Split School of Medicine, Šoltanska 2A, 21000 Split, Croatia; 9Department of Research, University Hospital of Split, Spinčićeva 1, 21000 Split, Croatia; ana.marusic@mefst.hr; 10Department of Research in Biomedicine and Health, Centre for Evidence-Based Medicine, University of Split School of Medicine, Šoltanska 2A, 21000 Split, Croatia

**Keywords:** hospital, code of ethics, ethics, medical, ethics committee, health personnel

## Abstract

**Background/Objectives**: Ethical challenges in healthcare require awareness and adherence to professional codes of ethics, particularly in interdisciplinary settings such as tertiary hospitals. This study aimed to assess the attitudes and awareness of healthcare professionals regarding codes of ethics at the University Hospital of Split in Croatia, which did not have an institutional code at the time of the study. **Methods**: A cross-sectional survey using a structured, anonymous questionnaire was distributed both physically across hospital departments and online via email. Welch’s *t*-test, ANOVA, Kruskal–Wallis, and correlation tests were used to assess associations between favourability scores and participant characteristics. Linear and logistic regression analyses further examined predictors of favourable attitudes. **Results**: Of 442 returned questionnaires, 377 were complete and included in the analysis, mainly from nurses (56.5%) and physicians (42.7%). The median favourability score was 83.8% (88/105; IQR 78.1–88.6), with 87.0% scoring above the favourable threshold (≥75%). Female gender and higher education were significantly associated with more favourable attitudes. Participants strongly endorsed core principles such as patient confidentiality and autonomy, yet 57.6% considered ethics education during training inadequate, and only 36.3% viewed dual practices as a conflict of interest. Most respondents reported adherence to ethical standards (85.4%), while only over half were familiar with their professional ethics code (64.5%) and the hospital Ethics Committee (56.2%); a total of 66.3% supported introducing a hospital-specific code. Awareness and support for ethical structures were higher among women and those with more education. **Conclusions**: This study reveals a gap between personal ethical commitment and institutional ethical infrastructure. Strengthening ethics education and implementing a hospital-specific ethics code may enhance organisational ethical culture.

## 1. Introduction

Normative ethics is usually represented as a code of conduct that integrates both perception of right and wrong and moral character [1]. A complex framework of healthcare ethics has been firmly established in the professionalism of healthcare workers around the four main principles of medical ethics—beneficence, nonmaleficence, autonomy, and justice [2,3,4]. The cornerstone of healthcare ethics comes from fundamental documents, such as the Hippocratic Oath [5], Nuremberg Code [6], and Helsinki Declaration [7]. While there are differences in views and knowledge of the code of ethics, medical professionals have moral, ethical, and legal obligations in their practice [8]. In clinical practice, the most reliable definition of healthcare ethics is identifying, analysing, and resolving moral dilemmas that impact patients while considering clinical facts within an interdisciplinary framework [9]. It can also be defined as regulations that guide physicians in their practice and prevent them from legal action [10]. Healthcare workers are regularly faced with moral dilemmas about different issues, including end-of-life care, resuscitation, competence, consent, care and treatment choices, and the general organisational administration of healthcare [11,12].

An essential quality for the practice of healthcare ethics is the knowledge of the professional code of ethics [13]. Although each healthcare profession traditionally upholds its own professional code of ethics, modern clinical practice increasingly relies on interprofessional collaboration in shared workplaces, such as hospitals, clinics, and long-term care institutions [14,15,16]. In such settings, ethical dilemmas often transcend the boundaries of individual professional roles [17]. For this reason, a common code of healthcare ethics among all healthcare professionals [18,19] is essential to healthcare professionals, and it serves to regulate professions and as a guide for moral behaviour in specific healthcare practices [20,21]. It does not replace profession-specific codes but rather complements them, providing a shared ethical foundation that fosters mutual understanding, coordinated decision-making, and consistent patient-centred care across disciplines [22].

The literature review on the practical use of professional codes in healthcare indicates that although most health professionals are aware of the existence of professional codes of conduct, they are often unfamiliar with their actual content and apply them infrequently in daily clinical practice, underscoring the need for codes that are effectively communicated and practically relevant in healthcare settings [21]. Also, studies from limited resource settings found that although doctors and nurses acknowledged the importance of ethical principles, they nevertheless perceived their ethics training as insufficient and demonstrated limited awareness of professional codes in daily practice [2,23].

Therefore, appropriate knowledge and a positive attitude toward the ethical code, as well as thorough training in medical ethics, allow healthcare professionals to anticipate, manage, decide over, and address ethical difficulties and challenges faced in daily practice [23].

In this study, we assessed the knowledge of and attitudes towards ethical principles of the employees in a university hospital in a public healthcare system. At the time of the study, the hospital did not have a uniform code of ethics, and its health professionals were guided by national professional codes, such as the ethics codes of medical and healthcare professional associations.

## 2. Materials and Methods

### 2.1. Study Design

This hospital-based cross-sectional study used a structured self-administered questionnaire about employees’ knowledge and attitudes towards codes of ethics. We reported our findings according to the Consensus-based Checklist for Reporting of Survey Studies (CROSS) [24].

### 2.2. Setting

The study was conducted at the University Hospital of Split in Croatia, a country which has a mandatory social health insurance system with almost universal population coverage [25].

The University Hospital of Split is a public tertiary-level hospital that offers highly specialised care, acts as a reference centre, engages in teaching and research, and provides services such as organ transplantation, cancer treatment, neonatal intensive care, cardiovascular surgery, and complex trauma. It is the largest hospital in the Split Dalmatia County and the second largest in Croatia, with over 4000 staff spread throughout 15 departments, 9 divisions, 7 sections, and sub-sections, 1500 acute and 30 chronic contractual beds, and 24 surgery rooms. Routine and administrative tasks are carried out by the directorate and twelve offices. The hospital treats over a million citizens of neighbouring Bosnia and Herzegovina and a million citizens of the Republic of Croatia, in addition to nearly half a million foreign visitors during the summer months [26].

### 2.3. Questionnaire Survey

Data were collected using a three-part structured self-administered questionnaire. The first part collected sociodemographic data on age, gender, work experience, and the participant’s current position in the hospital. The survey also asked respondents to indicate their highest educational qualification, defined as the entry-level degree required for professional practice (e.g., high school, bachelor’s, or master’s degree for most healthcare professions, and Doctor of Medicine for physicians, which, in Croatia, is obtained through a six-year integrated university program). In addition, participants were asked to report whether they had obtained a scientific degree, such as a Master of Science or PhD, which are awarded separately from professional qualifications. The second part consisted of twenty-one statements about codes of ethics, based on the Code of Ethics of the Croatian Medical Chamber, the Code of Ethics of the Croatian Chamber of Nurses, and other healthcare professionals, and questions from studies from other healthcare systems [23,27]; we developed the questions included in the questionnaire. Each survey question was assessed on a 5-point Likert scale, ranging from “strongly disagree” to “strongly agree.” The responses for each question were assigned numerical values (1 to 5), which were then aggregated to calculate the total score for each participant (maximum 105). Responses to negatively phrased items (3 questions in the survey) were inverted to compute the total score. A score of 75% or higher was considered a “positive favourability score,” indicating a generally favourable response across the survey [27]. This threshold was used to distinguish participants who exhibited a high level of positive favourability from those with lower favourability scores. The total score was used as the dependent variable in the linear regression models, reflecting the overall level of positive or negative favourability across all survey items.

The third part of the questionnaire consisted of four questions about the awareness of professional and hospital ethics codes and the hospital Ethics Committee, as well as an open statement about what they think should be in the hospital code of ethics.

### 2.4. Sample

We used simple convenience sampling of the employees at the University Hospital in Split present at their workplace at the times when the survey questionnaire was distributed (July to November 2024), making 3137 healthcare professionals employed at the time of the study eligible for inclusion. Based on a confidence level of 95% and a margin level of 5%, we aimed to include at least 343 healthcare professionals. The sample size was calculated using an online sample size calculator https://www.calculator.net/sample-size-calculator.html (accessed on 15 June 2024).

### 2.5. Survey Administration

We first distributed the physical versions of the questionnaire to all hospital departments and collected them after a few days in closed collection boxes to ensure anonymity. Then we distributed the questionnaire via the hospital administration, which sent an email containing the link to the online survey to all hospital employees. In the online version, we asked the employees not to complete the questionnaire if they had already filled out the paper version, in order to avoid duplicate responses. We did not send follow-up reminders.

### 2.6. Ethical Considerations

The University Hospital of Split Ethics Committee (Document Class: 520-03/24-01/136, Reg. No. 2181-147/01/06/LJ.Z.-24-02) provided the ethical approval for the study. We did not collect any personal data from the respondents. Both the electronic and physical questionnaires informed individuals about the survey, including the information that the online survey was programmed not to collect IP addresses to ensure full anonymity. The survey began with a notice informing individuals that, by continuing to answer the survey after being provided the information on the study, they consent to participate.

### 2.7. Statistical Analysis

Categorical variables are presented as counts and percentages, whereas continuous variables are summarised as medians with inter-quartile ranges (IQR) after assessment of normality with the Shapiro–Wilk test. Group differences in the total favourability score were evaluated with three tailored procedures: a Welch unequal-variance *t*-test for gender, one-way ANOVA for educational level (Bonferroni-adjusted post hoc contrasts), and a Kruskal–Wallis test for employment tenure (>10, 5–10, 1–4, <1 year). Associations with age were examined with both Pearson and Spearman correlation coefficients. For all these analyses, the original 5-point Likert scale data were used, while the aggregation into three categories was applied solely for descriptive presentation.

For the knowledge outcomes (“Yes/No/I don’t know”), Pearson’s χ^2^ tests were applied to every categorical predictor; Monte-Carlo *p*-values (10,000 replicates) or Fisher’s exact tests were used when ≥20% of expected counts were <5. Age was analysed with a Spearman rank test after coding the responses ordinally (No = 0, IDK = 1, Yes = 2). Cramer’s V was reported for significant χ^2^ results to convey effect size. In addition, separate single-predictor linear regressions were fitted for the continuous favourability score, and single-predictor logistic regressions (favourable ≥ 75% vs. <75%) were fitted for each participant characteristic; odds-ratio confidence intervals were obtained from the latter.

Two-sided *p*-values <0.05 were considered statistically significant. All analyses were performed in JASP Team (2024; Version 0.19.3); regression and supplementary χ^2^ diagnostics were cross-checked in Python 3.11 with the statsmodels 0.14 package.

## 3. Results

A total of 349 paper and 93 online questionnaires were collected. We excluded 65 incomplete responses (29 were online questionnaires), which left 377 questionnaires for analysis (Table 1). The majority of participants were female, and the median age of the respondents was 39 years (IQR 20–58). The cohort was predominantly composed of medical doctors (42.7%) and nursing staff (56.5%). Most of them worked directly with patients. Half of the participants had been employed at the hospital for more than 10 years.

The responses to a questionnaire aimed at assessing the ethical attitudes of healthcare professionals showed that healthcare professionals strongly supported core ethical principles in clinical practice (Table 2). The most favourable responses included near-universal agreement on respecting patient rights (96.3%), patient confidentiality and dignity (99.2%), and merit-based professional advancement (95%). Conversely, the least favourable responses reflected concerns about ethics education and system integrity. Furthermore, 37.4% expressed dissatisfaction with current ethical standards, and 57.6% found ethics education during training to be inadequate. Only 36.3% of respondents agreed with the statement that a healthcare professional employed in the public sector who also works in the private sector had a conflict of interest.

The median overall score was 83.8% of the maximum (88/105 points; IQR 78.1–88.6%). A total of 328 out of 377 respondents (87.0%) achieved a favourable attitude score—defined as ≥75% of the maximum 105 points (≥79 points)—while 49 participants (13.0%) fell below this threshold. The median total score was 70.5% (74/105; IQR 66.7–73.3%) in the low favourability group and 85.7% (90/105; IQR 81.0–88.6%) in the high favourability group.

Among the 377 respondents, women reported higher average favourability scores (83.7% ± 7.3) than men (80.8% ± 7.9) (Welch *t* test, *p* = 0.0011). Scores did not differ significantly by employment tenure overall (Kruskal–Wallis, *p* = 0.08), although staff with more than ten years of service showed the highest mean (83.6% ± 7.4), and those employed for five to ten years showed the lowest (81.2% ± 7.7). Educational level displayed a significant gradient (one-way ANOVA, *p* = 0.0039): participants holding a master’s degree achieved the highest mean score (85.6% ± 6.5), whereas high school graduates scored the lowest (80.9% ± 8.7). Age was not associated with favourability (Pearson r = 0.09; Spearman ρ = 0.08; both *p* > 0.07).

Separate single-predictor regressions were run for each participant characteristic, using the total favourability percentage as a continuous outcome (linear model) and the ≥75% threshold as a binary outcome (logistic model).

Gender showed the clearest effect. In the linear model, women scored, on average, 2.9 percentage points higher than men (overall F = 11.9, *p* = 0.0006). In the logistic model, male respondents were less than half as likely as females to reach the favourable threshold (OR = 0.42, 95% CI 0.22–0.77; LR χ^2^ = 7.5, *p* = 0.006).

Age did not contribute significantly in the linear model (β = +0.06 pp per year, *p* = 0.08). The corresponding odds ratio of 1.02 per year (*p* = 0.14) likewise failed to reach significance in the logistic analysis.

The hospital department accounted for a modest but significant share of the variance in the continuous score (overall F = 2.0, *p* = 0.0018). However, the same variable produced a singular matrix in the logistic model, suggesting that some departmental categories were too small or perfectly predicted the outcome.

The highest educational degree was also associated with favourability. The linear model detected an overall effect (F = 4.54, *p* = 0.0039), whereas the logistic model did not reach global significance (LR χ^2^ = 5.2, df = 3, *p* = 0.16). Individual contrasts (e.g., master’s vs. bachelor’s or high school) likewise remained non-significant after adjustment.

The professional field (medicine vs other) showed no association with favourability in either model (linear F = 1.48, *p* = 0.22; logistic *p* = 0.98). Scientific title, years employed in the hospital, and whether the respondent worked directly with patients could not be evaluated reliably: sparse or unbalanced categories led to unstable estimates and, in several cases, model non-convergence.

Taken together, these single-predictor analyses indicate that gender remains the strongest independent predictor of favourability, whereas age, clinical department, educational level, professional field, and the remaining employment factors do not show robust effects when examined in isolation.

With regard to their awareness of ethics codes and structures (Table 3), more than half of the respondents reported familiarity with their professional association’s ethical code, and about half were aware of the role of their institution’s Ethics Committee. Most participants (85.4%) stated they adhere to ethical standards in their work. Additionally, 66.3% supported the implementation of a specific institutional ethical code.

Associations between the three-level responses in Table 3 (“Yes”, “No”, “I don’t know”) and participant characteristics were examined with Pearson’s χ^2^ tests; age was analysed separately with a Spearman rank test after coding answers ordinally (No = 0, IDK = 1, Yes = 2).

Gender produced the clearest pattern. Women more often reported familiarity with their professional association’s ethics code (χ^2^ = 7.9, *p* = 0.019) and expressed stronger support for introducing a hospital-specific code (χ^2^ = 13.0, *p* = 0.0017). Educational attainment also mattered: awareness of the hospital Ethics Committee’s role differed by highest qualification (χ^2^ = 21.9, df = 6, *p* = 0.001; Cramer’s V = 0.17); master’s-level staff most frequently answered “Yes”, whereas high-school graduates were least aware.

Departmental affiliation showed only a modest influence. The association reached significance for code familiarity (χ^2^ = 30.6, df = 18, *p* = 0.012; V = 0.15) but was non-significant for the remaining three items (all χ^2^  *p* ≥ 0.08); logistic models could not be fitted because several small departments contained no “No” responses.

No other characteristic—age (Spearman ρ ≤ 0.07, *p* > 0.200, scientific title, length of service, degree field or day-to-day patient contact—showed a meaningful link to any knowledge outcome (all χ^2^  *p* ≥ 0.07; V ≤ 0.10, i.e., trivial-to-small effects).

Answering “Yes” to any of the knowledge items was consistently associated with a higher likelihood of holding an overall favourable ethical attitude (≥75% of the maximum score). Familiarity with the professional association’s code increased the odds of being favourable by a factor of 2.5 (OR = 2.53, 95% CI 1.38–4.66, *p* = 0.003), and knowing the role of the hospital Ethics Committee raised the odds 2.3-fold (OR = 2.26, 95% CI 1.22–4.18, *p* = 0.009). Self-reported adherence to professional ethical standards also predicted favourability, albeit less strongly (OR = 2.15, 95% CI 1.04–4.44, *p* = 0.039). The strongest relationship was observed among those who supported the creation of a dedicated hospital ethics code (OR = 2.80, 95% CI 1.52–5.15, *p* = 0.001).

Taken together, gender—and, for specific questions, educational level—were the main determinants of staff awareness of institutional ethics resources. Moreover, positive knowledge or attitudes toward such structures, whether professional, committee-based or policy-oriented, translate into markedly higher odds of an overall favourable ethical stance.

Regarding the open-ended survey question, which aimed to gather participants’ suggestions for the hospital code of ethics, no appropriate responses were provided. Consequently, no qualitative analysis could be conducted.

## 4. Discussion

Our study investigated the attitudes and awareness of healthcare professionals regarding codes of ethics at a large tertiary hospital in Croatia. We found a generally high level of ethical awareness and favourable attitudes among employees at the University Hospital of Split, with over 87% of respondents achieving a positive favourability score. Furthermore, while most healthcare professionals reported adhering to ethical standards and being familiar with their professional codes, only half of them knew the role of the hospital’s Ethics Committee. Also, the majority of respondents stated that the introduction of a hospital-specific code of ethics would be beneficial to their professional work, highlighting a general awareness of ethical frameworks and areas for improved institutional clarity and guidance.

The majority of respondents strongly endorsed core ethical values such as patient confidentiality, respect for patient autonomy, and merit-based professional advancement; however, many also perceived ethics education during their professional training as inadequate, expressed dissatisfaction with current ethical standards in practice, and, notably, only a third believed that dual employment in both the public and private sectors, known as dual practice, constitutes a conflict of interest. These findings highlight both the strengths and critical gaps in ethical alignment within the healthcare system. On one hand, the general acceptance of fundamental ethical principles shows that healthcare professionals have a strong foundation of individual moral commitment [28]. On the other hand, the evident dissatisfaction with ethics education and the lack of awareness of structural ethical challenges, such as conflicts of interest arising from dual practice, suggest inconsistencies between personal values and systemic ethical awareness [29]. This disparity may point to a lack of formal institutional guidance and a lack of conversation around applied ethics in complex organisational contexts [30], where, according to staff, the introduction of an institutional code of ethics is considered necessary, or it may reflect the existing legal framework that permits such dual practice, thereby normalising it within the professional culture [31].

The lack of consensus on issues such as dual practice has also been documented in international literature, particularly in health systems facing transitional or resource-related challenges [32,33]. The most common justification for physician dual practice is low public-sector salaries, particularly in resource-limited settings [33]. While that practice is more widespread in low- and middle-income countries, which tend to have larger populations, weaker health systems, higher out-of-pocket costs, and lower physician density, the primary negative consequence remains reduced quality of care in public hospitals [33]. In Croatia, this has led to patients being more willing to pay for faster access to higher-quality care, contributing to the growth of private practices and a rising number of physicians working in the private sector [34]. To respond to this problem, the Ministry of Health has proposed a regulation to restrict dual practice by allowing public sector doctors to work privately only if patients wait less than 60 days for a first consultation, aiming to improve healthcare accessibility and prevent excessive delays in the public system, where the average waiting time for diagnostic tests is currently 126 days [35]. These findings support stronger regulatory frameworks, improved public-sector compensation, and enhanced monitoring systems to mitigate absenteeism, prevent directing patients to private practices, and ensure equitable care delivery in public health systems [33,36].

We also found that gender and education were significant factors influencing ethical attitudes and awareness. Female respondents not only scored higher on favourability but were also more familiar with professional ethical codes and more supportive of introducing a hospital-specific code. Likewise, participants with higher education, especially those with a master’s degree, showed greater awareness of institutional ethical structures compared to those with lower qualifications. These patterns highlight the role of both gender and education in shaping ethical engagement and underline the need for stronger ethics education and institutional Ethics Committee visibility [37].

When compared to other studies, our results have some similarities and differences with findings from similar healthcare systems in transition or resource-constrained contexts. A study conducted in Addis Ababa, Ethiopia [27] found that while many were aware of the existence of the Ethiopian Health Professionals Code of Ethics, awareness of their Ethics Committee was notably lower, and among those familiar with the Committee, most did not know its specific powers and responsibilities. These findings are similar to those of a study conducted among residents in Pakistan [38], where attitudes among respondents were mostly positive, but more than 80% of respondents were not aware of the institutional Ethics Committee. Additionally, the study conducted in Barbados [39] revealed limited awareness of the hospital’s Ethics Committee among doctors and nurses, with a notable proportion of physicians perceiving the committee as ineffective. Although Croatia, met the legal requirements for establishing Ethics Committees by the 1997 “Law on the Health Protection” which obliges all healthcare institutions in Croatia to have Ethics Committees whose purpose is to provide ethics consultations, support healthcare professional in resolving ethical issues during clinical practice and analyse research protocols [40], our study confirmed that in practice their role is often limited, as healthcare professionals are unfamiliar with their intended functions and rarely turn to them for support [41].

With regard to the Central and Eastern European context, no large-scale quantitative studies have been conducted that comprehensively examine the attitudes of all healthcare professionals. Existing research is limited to specific groups or national settings. For example, a study among nurses in six European countries found that many were unfamiliar with the content of professional codes of ethics and rarely applied them in daily practice, often relying instead on personal experience or the prevailing organisational culture [42]. Also, the first national survey in Slovenia among healthcare professionals from 14 hospitals showed that, although ethical dilemmas were frequently encountered, hospital Ethics Committees were rarely consulted, indicating limited awareness and underutilisation of these Committees [43]. These findings suggest that, while there is evidence of ethical and organisational challenges in Central and Eastern Europe, further research is needed to provide a broader and more systematic understanding of healthcare professionals’ perspectives across the region.

To our knowledge, this is the first study in Croatia, and among the first of its kind in Europe, to assess the attitudes and awareness of healthcare professionals regarding ethical codes in a tertiary hospital setting. Our results suggest that, although healthcare professionals expressed a strong personal commitment to ethical values, their limited familiarity with institutional Ethics Committees, the absence of a unified ethical code, and the lack of consensus on issues such as physician dual practice all indicate structural and organisational gaps that hinder the translation of individual ethical attitudes into practice. Several mechanisms may help explain these findings. For instance, hospital Ethics Committees in many Central and Eastern European countries have traditionally been established more as formal requirements than as active consultative bodies, which may leave healthcare staff insufficiently prepared or motivated to engage with them in resolving ethical dilemmas [44]. Also, systemic challenges such as resource constraints, workforce shortages, and competing priorities in daily clinical work may reduce the perceived practicality of seeking institutional ethics support [45]. In our opinion, in addition to these structural and systemic factors, a lack of compliance with ethical codes may also be influenced by individual-level determinants, including insufficient personal ethical sensitivity, limited knowledge of ethical principles and codes, as well as shortcomings in undergraduate or postgraduate curricula.

The strength of this study lies in its robust sample size and inclusion of diverse healthcare professionals, allowing for a comprehensive assessment of ethical attitudes across staff categories. Also, the questionnaire was adapted from previously validated tools and contextualised for the Croatian healthcare environment. Our study also has some limitations. The study was conducted in a single institution, which may affect generalizability. However, as most hospitals in Croatia are within the public healthcare system, these findings are applicable to the whole country and possibly also to public healthcare systems in countries with similar socioeconomic and political history, such as those in Central and Eastern Europe. As with similar studies, social desirability bias may have influenced responses, as participants may have answered based on perceived expectations rather than personal convictions. We tried to mitigate this by fully anonymising the survey. As we distributed the survey in two formats (online and paper), it is possible that some participants answered twice; however, we believed that this is not likely for busy hospital employees, especially as they have been warned not to fill in the physical questionnaire if they had completed the online one, and vice versa. In addition, as the online survey link was distributed via hospital administration, this mode of distribution may have introduced some bias, given the personal nature of the questions. We chose this method, together with the paper format, to maximise the response rate, since it was not feasible to personally visit and supervise questionnaire distribution across all departments. Nevertheless, we emphasised anonymity and clearly stated that no identifying information, including IP addresses, would be collected, which we believe helped to minimise any potential influence of distribution via management. The use of convenience sampling may have introduced selection bias, and the cross-sectional design limits causal inference.

## 5. Conclusions

Beyond its national relevance, our study contributes to the more expansive discussion on the integration of ethics in healthcare systems by highlighting a crucial gap between individual ethical attitude and institutional ethical infrastructure. While respondents expressed strong personal commitment to ethical values and professional codes, their limited knowledge of the hospital’s Ethics Committee and the lack of consensus on issues such as physician dual practice reflect a disconnect between ethical ideals and practical realities. This gap is especially relevant for health systems running under a universal health coverage system, like Croatia [25], where ethical integrity is crucial for maintaining equity, transparency, and confidence.

The findings may be relevant to policymakers and hospital administrators seeking to enhance organisational ethics, clinical quality, and accountability in similar healthcare systems. The universal support for a hospital-specific code of ethics, which was not implemented at the time of study, indicates that healthcare professionals are seeking a clearer, cohesive, and contextually relevant ethical framework that addresses the complexities of interprofessional collaboration and routine challenges in clinical work. In contrast to universal professional codes, a locally formulated institutional code can align expectations among various healthcare roles, clarify procedures in ethically confusing circumstances, and direct staff when individual codes may be insufficient or when institutional values require interpretation in the context of local practice. An effectively implemented code could strengthen the function of the hospital’s Ethics Committee by explicitly specifying its responsibilities, increasing transparency, and promoting staff involvement.

Given that higher education and female gender were consistently associated with more favourable ethical attitudes, ethics-related interventions should consider both demographic and professional dynamics. Tailored ethics training, differentiated by professional group, could help address the unique ethical challenges faced by physicians, nurses, and non-clinical staff alike.

Importantly, the findings underscore that fostering an ethical organisational culture requires more than individual commitment; systemic changes are needed at both the institutional and policy levels. Structural changes such as introducing hospital-specific codes of ethics, strengthening the role of hospital Ethics Committees, improving ethics education, and incorporating routine assessments of ethical awareness into quality improvement initiatives could bridge the gap between individual ethical commitment and institutional support, thereby fostering sustained ethical development and enhancing the overall quality of care.

Future research should aim to validate and expand the use of our instrument in broader, multicentre studies across Central and Eastern Europe, enabling regional comparisons and the identification of shared and country-specific challenges. Comparative studies between different types of healthcare sectors (primary and tertiary or public and private) could shed light on organisational factors that influence ethical awareness and practice. It would also be valuable to investigate how the establishment or strengthening of hospital Ethics Committees and the implementation of hospital-specific codes of ethics affect the resolution of ethical dilemmas in practice. Moreover, intervention studies could evaluate the impact of targeted ethics training on healthcare professionals’ knowledge, confidence, and willingness to use institutional mechanisms. Finally, future research could use qualitative methods, such as focus groups, to gain deeper insights into ethical decision-making that cannot be fully captured by surveys alone.

## Figures and Tables

**Table 1 healthcare-13-02131-t001:** Participants’ demographic characteristics.

Gender	No. (%)
Male	105 (27.9)
Female	272 (72.1)
**Age in years, median (inter-quartile range)**	39 (31–50)
**Highest educational qualification:**	
High school	67 (17.8)
Bachelor’s degree	101 (26.8)
Master’s degree	48 (12.7)
Doctor of medicine	161 (42.7)
**Years of employment:**	
<1 year	28 (7.4)
1–4	85 (22.5)
5–10	73 (19.4)
>10 years	191 (50.7)
**Type of working staff:**	
Medical doctor	161 (42.7)
Nursing staff	213 (56.5)
Physical therapy staff	3 (0.8)
**Working with patients:**	
Yes	345 (91.5)
No	31 (8.2)
**Scientific degree:**	
Doctoral degree (PhD)	39 (10.3)
Master of science degree	9 (2.4)
No scientific degree	329 (87.3)

**Table 2 healthcare-13-02131-t002:** Responses to questions related to attitude towards medical ethics.

Statements:	Strongly Disagree or Disagree, No. (%)	Neither Agree Nor Disagree, No. (%)	Agree or Strongly Agree, No. (%)
As a healthcare professional, you will respect the right of a mentally competent and conscious patient to make an informed and voluntary decision to accept or refuse a particular physician or recommended medical assistance.	3 (0.8)	11 (2.9)	363 (96.3)
Healthcare professionals must respect the confidentiality, privacy, choices, and dignity of the patient.	1 (0.3)	2 (0.5)	374 (99.2)
Healthcare professionals are permitted to provide healthcare services that do not address the needs of the patient but rather serve their own personal benefit. *	315 (83.5)	38 (10.1)	24 (6.4)
A healthcare professional may use scientifically unverified methods for treatment and health promotion and collaborate with those who misuse public trust by promoting such methods. *	351 (93.1)	10 (2.65)	16 (4.2)
Every healthcare professional must refuse any gift or service that could be interpreted as an attempt to gain personal advantage or benefit.	77 (20.4)	72 (19.1)	228 (60.5)
Criteria for evaluating and advancing in the medical profession should be based on expertise, ability, professional merits, and work results.	8 (2.1)	11 (2.9)	358 (95)
In dealing with patients, the physician will act economically and in accordance with the principles of rational medical practice. They will not perform unnecessary examinations and treatments, regardless of who bears the healthcare costs.	18 (4.8)	51 (13.5)	308 (81.7)
Every healthcare professional must use medical equipment or health technology that has been scientifically proven to meet the claims made about it.	11 (2.9)	48 (12.7)	318 (84.4)
A healthcare professional has the right to exercise conscientious objection, provided it does not result in permanent harm to the patient’s health or endanger the patient’s life.	25 (4.5)	35 (9.3)	317 (84.1)
After exercising conscientious objection, the healthcare professional must promptly inform their superiors and the patient, directing the patient to another colleague of the same profession.	11 (2.9)	33 (8.8)	333 (88.3)
Continuing intensive treatment for a patient in the terminal phase of a disease is unjustifiable and deprives the dying person of their right to a dignified death.	63 (16.7)	76 (20.2)	238 (63.1)
A healthcare professional may refuse further care and refer a patient to another colleague if, despite adequate information and decision-making capacity, the patient does not follow treatment and disease prevention recommendations, provided the patient’s life is not acutely endangered.	50 (13.3)	91 (24.1)	236 (62.6)
A physician may prescribe medications or formulations without knowledge of their composition or pharmacological effects. *	234 (88.6)	29 (7.7)	14 (3.7)
In the case of brain death, determined by a professionally accepted method, the physician is obligated, under relevant legislation, to maintain the life of organs, body parts, or tissues that may be used for the treatment of other patients.	8 (2.1)	56 (14.9)	313 (83)
Every healthcare professional must report unethical, immoral, or medically inappropriate conduct of a colleague to the relevant authority.	18 (4.8)	69 (18.3)	290 (76.9)
Every healthcare professional must report their own mistakes to the relevant authority.	32 (8.5)	68 (18)	277 (73.5)
If the treatment needs of a patient exceed your capabilities, knowledge, or skills, you are required to refer the patient to another healthcare professional who can fulfill those needs and is obligated to respond to such a request.	8 (2.1)	14 (3.7)	355 (94.2)
A healthcare professional employed in the public sector who also works in the private sector has a conflict of interest.	137 (36.3)	103 (27.3)	137 (36.3)
The current level of ethical medical practice among healthcare professionals is satisfactory.	141 (37.4)	150 (39.8)	86 (22.8)
In my opinion, the level of medical ethics education during academic training is inadequate.	59 (11.7)	100 (42.3)	217 (57.6)
Education in medical ethics is essential for all healthcare professionals.	13 (3.4)	29 (7.7)	335 (91.5)

* Negatively phrased questions whose score was inverted to calculate the total score.

**Table 3 healthcare-13-02131-t003:** Knowledge about the hospital code of ethics and the work of the hospital Ethics Committee, adherence to ethical standard and opinion about a separate hospital ethics code (number, %).

Question	Yes No. (%)	No No. (%)	I Do Not Know No. (%)
Are you familiar with the Ethics Code of your professional association?	243 (64.5)	62 (16.4)	72 (19.1)
Do you know the role of the Ethics Committee in your institution?	212 (56.2)	74 (19.6)	91 (24.1)
Do you adhere to the ethical standards of your profession in your work?	322 (85.4)	4 (1.1)	51 (13.5)
Do you believe that introducing a specific Ethical Code in the hospital would be beneficial to your work?	250 (66.3)	30 (8.0)	97 (25.7)

## Data Availability

The study data will be available upon reasonable request to the corresponding author.

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
