# Peer review of "Understanding Employees’ Attitudes and Awareness of Code of Ethics and Associated Factors: A Cross-Sectional Survey at a Public Tertiary Hospital in Croatia"

_healthcare, 2025, doi:10.3390/healthcare13172131_

Round 1

Reviewer 1 Report

Comments and Suggestions for Authors

I would like to thank you for sending the article for review and for the opportunity to refer to its content. The article is not technically correct. The article presents the background and significance of the problem, but lacks: indication of the research gap, the purpose of the article and research hypotheses and contribution of work, e.g. indication that the study brings added value through a survey conducted in a hospital employing a large number of employees.

  1. Please complete the introduction.

The paper lacks a literature review on the assessment of knowledge and attitudes towards ethical principles and including aspects included in the survey questions.

  1. In the literature review, please indicate the research in other countries of exactly the topics covered by the survey, and do not omit any important thematic block.

In the discussion, line 330 presents citations to studies in Ethiopia, Pakistan and Barbados, and states (lines 356-357) that the assessment may apply to all countries of Central and Eastern Europe. This statement is not supported by research or literature review.

  1. In the discussion, please explain what the results mean and what mechanisms may explain them, and to relate the results to previous studies by other authors of Central and Eastern Europe.
  2. Please give suggestions on what could be investigated in the future using developed research methods to develop the problem undertaken.

Reviewer 2 Report

Comments and Suggestions for Authors

Dear Editor and Authors,

Thank you for the opportunity to review your manuscript entitled “Understanding employees’ attitudes and awareness of code of ethics and associated factors: a cross-sectional survey at a public health care tertiary hospital in Croatia”

The topic addressed is both pertinent and valuable, particularly in the context of rural healthcare delivery.

Title: Clear and appropriate.

Keywords: “Healthcare professionals” and “Croatia” are not MeSH terms. I suggest replacing the former with “Health Personnel.”Abstract: Concise and well structured.

Abstract: Concise and well structured.

Introduction: Effectively contextualises the importance of ethics in hospital practice. Cites relevant literature and foundational documents.

Methodology

  • How was the open-ended question analysed?
  • Page 3, lines 133–134: the hyperlink provided does not work.
  • Survey administration states that questionnaires were distributed in paper format and later via email through hospital administration. Why were these two methods used? Given the personal nature of the questions, could distribution via management have influenced responses?

Results

  • In Table 1, were all the professionals surveyed medically qualified doctors? Under “highest education qualification,” general academic degrees are listed—why specify “Doctor of Medicine”? What is the distinction between “highest education qualification” and “scientific degree”? The overlap between professional and academic titles could be clarified.
  • Although the questionnaire was assessed using a 5-point Likert scale, Table 2 presents aggregated responses in three categories. It would be helpful for the authors to clarify whether this grouping was applied solely for presentation purposes or also used in statistical analyses. A brief rationale for this aggregation would enhance transparency and interpretability.

Discussion: Interprets the results well in light of international literature. Limitations are acknowledged. The authors note as a limitation the possibility that some participants may have responded twice, which could introduce significant bias into the study’s data.

Conclusion: How might these findings influence practice in terms of structural changes?

References: Appropriate and relevant.
